# UV Protection in the Cornea: Failure and Rescue

**DOI:** 10.3390/biology11020278

**Published:** 2022-02-10

**Authors:** Thomas Volatier, Björn Schumacher, Claus Cursiefen, Maria Notara

**Affiliations:** 1Department of Ophthalmology, Faculty of Medicine and University Hospital Cologne, University of Cologne, 62, 50937 Cologne, Germany; claus.cursiefen@uk-koeln.de (C.C.); maria.notara@uk-koeln.de (M.N.); 2Cologne Excellence Cluster for Cellular Stress Responses, Aging-Associated Diseases (CECAD) and Center for Molecular Medicine (CMMC), University of Cologne, Joseph-Stelzmann-Strasse 26, 50931 Cologne, Germany; bjoern.schumacher@uk-koeln.de; 3Center for Molecular Medicine Cologne (CMMC), Faculty of Medicine and University Hospital Cologne, University of Cologne, 21, 50931 Cologne, Germany

**Keywords:** cornea, UV, autophagy, nucleotide excision repair

## Abstract

**Simple Summary:**

The sun is a deadly laser, and its damaging rays harm exposed tissues such as our skin and eyes. The skin’s protection and repair mechanisms are well understood and utilized in therapeutic approaches while the eye lacks such complete understanding of its defenses and therefore often lacks therapeutic support in most cases. The aim here was to document the similarities and differences between the two tissues as well as understand where current research stands on ocular, particularly corneal, ultraviolet protection. The objective is to identify what mechanisms may be best suited for future investigation and valuable therapeutic approaches.

**Abstract:**

Ultraviolet (UV) irradiation induces DNA lesions in all directly exposed tissues. In the human body, two tissues are chronically exposed to UV: the skin and the cornea. The most frequent UV-induced DNA lesions are cyclobutane pyrimidine dimers (CPDs) that can lead to apoptosis or induce tumorigenesis. Lacking the protective pigmentation of the skin, the transparent cornea is particularly dependent on nucleotide excision repair (NER) to remove UV-induced DNA lesions. The DNA damage response also triggers intracellular autophagy mechanisms to remove damaged material in the cornea; these mechanisms are poorly understood despite their noted involvement in UV-related diseases. Therapeutic solutions involving xenogenic DNA-repair enzymes such as T4 endonuclease V or photolyases exist and are widely distributed for dermatological use. The corneal field lacks a similar set of tools to address DNA-lesions in photovulnerable patients, such as those with genetic disorders or recently transplanted tissue.

## 1. Introduction

### The Cornea

The cornea is the superficial shield at the front of the eye. Its transparency is essential for the transmission of light into the eye and through to the retina, enabling visual perception. The cornea functions as a physical barrier, protecting the inner contents of the eye, while also providing a significant portion of the refraction needed for proper vision. The human cornea consists of five distinct layers. The anterior layer is the epithelium, and its underlying fibrous mesh is called the Bowman layer [1]. Posterior to the Bowman’s layer is the collagen-rich stroma, contributing to approximately 90% of the total corneal thickness. Beneath the stroma is Descemet’s membrane, which separates stroma from the most posterior corneal layer, the single cell layer endothelium (see Figure 1) [2,3]. The corneal structure does vary between species, particularly in terms of Bowman’s layer and Descemet’s membranes, to accommodate different needs and adaptations, although all fulfil the basic roles of protection and refraction [4].

The average radius of the anterior corneal surface is 15 mm and the average diameter of the posterior corneal surface is 13 mm. The shape of a healthy human cornea will change with age, typically thinning at the periphery [5].

The corneal epithelium covers the anterior portion of the cornea; it is composed of five to six layers of stratified, noncornified, squamous cells. The epithelium’s basal layer of cuboidal cells adheres to the Bowman layer via a basement membrane [6]. The main purpose of the corneal epithelium is to form a high turnover barrier to the outside. Along with this protective role, the epithelium also produces cytokines that influence the behaviour of neighbouring cell types, such as keratocytes [7]. The smaller basal cells are progenitor cells that replenish the continuously depleted upper layers. The basal progenitors themselves are a population maintained by, and descended from, limbal epithelial stem cells [1]. Inside the limbus are radially oriented fibrous folds called the palisades of Vogt; the limbal stem cells reside within these ridges [8]. As the limbus acts as the interface between the vascularized conjunctiva and avascular cornea, the palisades themselves are partially vascularized. The vasculature is responsible for providing nutrition and oxygen to the limbal area [9]. It is currently thought that the limbus and its inhabiting cells maintain the border between the two different tissues, and that any damage to the limbus may enable an invasion of conjunctival cells into the cornea. This typically leads to vascularization and a loss in vision quality [10,11,12]. These disturbances may occur following UV damage where the limbal stem cells no longer maintain the border [13,14,15,16]. Although the role of epithelial factors in the maintenance of angiogenic and lymphangiogenic privilege seems evident, the role of the limbus as a physical barrier to vascular invasion seems poorly supported [17,18].

Damage to the epithelium must be repaired rapidly to restore barrier function and protect the cornea from bacteria or further trauma [19]. As the epithelial cell population is maintained by limbal stem cells (see Figure 2), the closure of wounds is severely impeded by any damage to their niche, the limbus [20]. The limbus can be considered a particularly critical component of the cornea as this perimeter of crypts is essential in maintaining a clear cornea fully covered in epithelium. While the limbus is vulnerable to threats ranging from viral and bacterial to chemical and traumatic, one particular source of stress is almost always present: UV radiation.

## 2. UV Effects on the Cornea

### 2.1. UV Damage and Repair Mechanisms

There are only two surfaces of the human body that are chronically exposed to UV: the skin and the eye [21]. The effects of both acute and chronic UV exposure on the skin are well documented [22,23,24,25], with several investigations into each of the cell types, tissues, and layers of the skin. In the case of the cornea, acute UV exposure is known to cause photokeratitis, a painful inflammation where the epithelium, stroma, and endothelium may be affected and which leads to clouding [26,27,28,29]. Chronic UV exposure usually leads to long term conditions such as tumours (squamous cell carcinomas, malignant melanomas, lymphoma of the conjunctiva [30,31,32]) or keratopathy (see Table 1) [33,34,35].

UV radiation entering the temporal limbus has been shown to focus at the nasal limbus, damaging limbal cells to a greater degree. This occurs specifically with UV entering at the temporal limbus that is then concentrated to the nasal limbus (see Figure 3) [36]. These damaged limbal cells and their descendants may then migrate from the limbus towards the center of the cornea [37].

### 2.2. Pterygium Aetiology and Pathogenesis

One of the first clear definitions of pterygium describes a pterygos, a wing-shaped degenerative and hyperplastic process where the bulbar conjunctiva invades onto the cornea [66]. These vascularized, fibrotic degenerations continuously advance across the cornea over time [67]. It is well documented across several ethnicities, locations, and age-groups that the primary risk factor of pterygia is UV [68,69,70,71,72,73]. The incidence is generally higher for men than for women (14.5% vs. 13.6%), and unilateral cases were more prevalent than bilateral cases [74]. The prevalence of pterygium increases with age, with the age group 55–59 showing the greatest vulnerability [75]. Pterygia are typically found at the interpalpebral zone of the cornea. Due to the peripheral light focusing effect, they will develop more often at nasal side and less often on the temporal side of the cornea [76].

### 2.3. UV-Induced DNA Lesion Formation

UV irradiation damages cellular DNA by causing the formation of cyclobutane pyrimidine dimers (CPDs) and pyrimidine (6-4) pyrimidone photoproducts (6-4PPs) (see Figure 4) [77]. CPD and 6-4PP lesions are both results of neighbouring pyrimidine covalently binding to each other [78,79,80]. CPDs are the most frequent UV lesions and—when cells fail to induce cell death through the activation of the DNA damage response—lead to mutations that in turn promote malignancy [81,82,83]. Mutations amid CPDs are responsible for transition mutations via Cytosine and Thymine nucleotide dimerization [84,85]. Importantly, CPD lesions are estimated to occur three times as frequently as 6-4PP lesions, and, due to their greater steric aberration in the DNA, 6-4PPs, they are more effectively recognized and thus repaired by NER than CPD lesions [86]. The first DNA repair enzymes that were discovered were CPD- and 6-4PP-photolyases, which specifically bind to either a CPD or a 6-4PP, respectively. These cryptochrome enzymes absorb visible (blue) light as energy source for electron transfer to mend the lesions [87,88,89]. In the Sphingomonas Antarctic bacterium a photolyase capable of binding to both CPD and 6-4PP called UV9 was identified [90,91].

Photolyases exist in organisms ranging from bacteria to plants and marsupials but are absent in placental mammals. Humans in fact lack CPD and 6-4PP photolyases and instead rely on nuclear excision repair (NER) pathway [92,93]. NER is far more versatile in removing a range of helix distorting lesions; however, they are far less effective to recognize CPD and 6-4PP lesions than the photolyases that have been selected to bind to only those lesions with high specificity. In contrast to the single enzyme photolyases, NER is executed by a highly complex mechanism involving dozens of proteins. While these proteins will not necessarily target UV damage specifically like the aforementioned photolyases, they are capable of repairing CPD and 6-4PP [94,95]. NER is initiated by lesion recognition by either the Xeroderma pigmentosum complementation group C (XPC) acting in concert with the RAD23 homolog B (RAD23B) as part of global genome (GG-) NER or by the Cockayne syndrome B (CSB) protein encoded by the excision repair cross complementation group 6 (ERCC6) gene as part of transcription-coupled (TC-) NER, dependent on the location of the damage. TC-NER and GG-NER differ in the way they recognise DNA damage. GG-NER relies on the XPC complex constantly probing the DNA for lesions while TC-NER relies on RNA polymerase stalling at lesion sites to recognize damaged DNA [96].

Both GG- and TC-NER recruit the NER core machinery including XPA and the TFIIH complex that unwind the double helix, verify the lesion, and subsequently recruit the ERCC1-XPF and the XPG endonucleases to excise a stretch containing the lesion [97]. Mutations in NER genes typically result in UV hypersensitivity of the skin and particularly when GG-NER is affected in a several thousand fold higher incidence of skin cancer in Xeroderma pigmentosum (XP) patients. In contrast, mutations affecting TC-NER result in Cockayne syndrome (CS) that is characterized by growth retardation and premature aging but not skin cancer. Distinct mutations in the TFIIH components XPD can lead to XP, rare combinations of XP and CS as well as trichothiodystrophy (TTD) that shares many features with CS but in addition leads to transcription elongation defects that cause brittle hair and nails [98,99]. These three diseases can also present eye-related anomalies such as conjunctivitis, photophobia, keratitis, pterygium, and corneal opacity [100], although these symptoms do not manifest in all patients.

One of the ways in which skin epithelia and corneal epithelia differ is in their incidence of UV-caused pathologies. Ocular surface pathologies such as pterygium, intraepithelial neoplasia, or carcinoma, have a lower rate of incidence than their equivalent in the skin (including melanomas) [101,102]. It should be noted that these mutation-driven pathologies are not always driven by UV but can also be due to viral factors. Human papillomavirus in particular is a source of mutations [103]. However, it is well known that there is a clear correlation between UV exposure and the occurrence of ocular surface pathologies, with particularly high incidences occurring in populations that live within 30 degrees latitude of the equator where UV radiation is high [104,105]. Furthermore, lesions occur more often in the sun-exposed interpalpebral fissure, specifically in the nasal or temporal regions within the limbus [106]. The eyes are certainly one of the areas most vulnerable to UV light, with an estimated 5 to 10% of all skin cancers occurring in the eyelids [45]. Even so, with all this evidence that UV does drive mutation and cancerous lesions in the cornea, the corneal epithelia is orders of magnitude less prone to UV-induced cancer than skin [107].

### 2.4. The Role of Genotoxic Stress

The reason for the cornea’s greater resistance to UV lies in its ability to deal with genotoxic stress. Specifically, components of the genotoxic stress detection and response cascade such as DNA damage-binding protein 2 (DDB2), XPC, and tumour suppressor p53 are present in greater amounts in corneal epithelial cells than in epidermal keratinocyte [108]. The quantity of a protein within a cell is a result of the balance between production and degradation. In the corneal epithelial cells, the degradation of XPC, DDB2, and p53 is slowed such that the active forms of these proteins are more common [108]. While it seems like the cornea is better equipped than the skin to deal with UV damage, this could be interpreted as a compensation measure because the skin has corneocytes that form a physical barrier and prevent some genotoxic stress from occurring in the first place. The cornea instead relies on the tear film, which does not present as effective a barrier to UV [109,110]. Additionally, the entire skin contains melanocytes that distribute melanin to adjacent cells via melanosomes; this protects nuclear DNA from UV [111,112]. In the corneal limbus, dendritic melanocytes reach out to surrounding cells and provide a similar protection [113,114]. When epithelial cells from skin and cornea are cultured without cornified layers, tear film, or melanin to protect them, the corneal cells seem to survive UV exposure for longer due to their more efficient photolesion repair. UV-induced apoptosis is typically triggered via detection of a critical quantity of DNA damage [108,115]. While both corneal and dermal epithelial cells are equally prone to UV-induced CPD formation, corneal cells survive best in vitro owing to their greater efficiency in repairing mutagenic DNA damage while dermal cells perish faster due to their slower rate of DNA damage repair [116,117].

### 2.5. Reactive Oxygen Species

UV may also damage epithelium by causing the formation of reactive oxygen species (ROS), free radicals that may oxidise cellular material such as proteins, organelles, or DNA [118]. When the amount of ROS in a cell exceeds the cellular antioxidant levels, damage will occur more readily and the cells may undergo apoptosis [119]. Endogenous antioxidants such as catalase, glutathione peroxidase (GPx), superoxide dismutase (SOD), glutathione (GSH), and scavengers, like uric acid, coenzyme Q, and lipoic acid are the principal defence against ROS [120]. When oxidative damage occur in a cell, the damaged element is then repaired or marked for elimination by the appropriate enzyme. In the case of nucleic acid damage, there are specialised enzymes, damaged proteins are removed from circulation by proteolytic systems, and damaged lipids are repaired by phospholipases, peroxidases, and acyl transferases [121,122]. ROS may damage any of the four bases of DNA and transform the functional base into any one of several potential lesions [123]. One of these lesions, 8-Oxoguanine (8-OxoG) in particular is linked to greater cancer cell survival [124]. The base excision repair (BER) process that repairs such damage begins with detecting the lesion via DNA glycosylases, in the case of 8-OxoG, Oxoguanine glycosylase 1 (OGG1) binds to the lesion [125]. The DNA glycosylase then cleave the oxidised base out of the DNA, causing a single-strand break. These breaks are detected by PARP proteins that act as the sensor molecules for these single-strand breaks in order to guide gap-filling complexes to repair the break and prevent cell death [126,127].

There are specific cellular repair mechanisms such as CPD being repaired by NER and ROS damage being repaired by base excision repair (BER). There are also cell functions that reduce the effect of UV damage that are not expressly there to mitigate such damage. In the case of cell proliferation, cell division allows for “dilution” of the damage to descendent cells [128]. The high turnover of epithelial tissue is a protective feature shared by both the skin and the cornea. The obvious point of failure in that system is the progenitor population; in the case of the cornea, that would be the basal limbal epithelial cells [129].

Thus, there exist multiple ways by which the corneal epithelial cells may protect themselves from UV, repair UV-caused damage, and remove damaged cells. However, each of these mechanisms can fail and when they do, UV becomes more reliably pathogenic.

### 2.6. Disruption of Autophagy Mechanisms

Autophagy is a degradation mechanism that utilises the lysosome to remove proteins from the cell. This is done either as part of the natural turnover of proteins or as part of a stress response. The process is mediated by several regulators that transport the targeted proteins and control the rate at which the proteins are removed. This rate can be adjusted by several stimuli to remove damaged proteins. One of these stimuli is UV irradiation. Ataxia-telangiectasia mutated (ATM) and Rad3-related (ATR) are both damage sensors that can mediate the activation of p53 via checkpoint kinase 1 (CHK1) and checkpoint kinase 2 (CHK2). ATR may also activate the STK11/AMPK metabolic pathway to stimulate the tuberous sclerosis 2 (TSC2) tumor suppressor and regulate autophagy via Beclin 1 [130,131]. TSC1 and TSC2 may also downregulate mTOR activity [132]. The difference between ATM and ATR is that the former is an ATM-Chk2 pathway mostly activated by double-strand breaks, while the latter ATR/Chk1 is activated by single-strand breaks [133,134]. Both ATR and ATM inactivate mTORC1, enabling autophagy as a result [135].

UV has also been found to stabilize p53, enabling it to be one of the starting points of the cellular response to UV stress by greatly increasing the amount of p53 within the cell [136]. In enucleated cells, the inhibition of p53 leads to increased autophagy, indicating that the cytoplasmic p53 regulates autophagy [137]. Despite evidence that p53 activation upregulates autophagy [132,138,139,140], it has also been observed that deletion of p53 may induce autophagy, and this was rescued with cytoplasmic, not nuclear, p53 [141]. Normally, p53 is localized in the cytoplasm of corneal epithelial cell. Following UV exposure, p53 expression increases and the cytoplasmic p53 migrates to the mitochondria [142]. There is a complexity to the way that p53 regulates autophagy, as it may be an inhibitor or an activator depending on the stimuli [143].

Another member of the p53 family, p73, is also linked to the regulation of autophagy. It has been documented that p73 represses the expression of autophagy-associated UV irradiation resistance-associated gene (UVRAG) [144]. UVRAG is a positive regulator of the Beclin1–PI(3)KC3 complex which itself induces autophagy [145]. 

Autophagy is a continuous process with several regulating factors. Following the initial formation of the early phagosome, The LC3 will begin separately to finalize the phagosome formation and allow it to be trafficked to initiate the process of lysosomal breakdown. This process involves the Atg10/Atg7-mediated process of Atg5-Atg12-Atg16 conjugation [146]. The sirtuin family has been documented as regulator of autophagy via this LC3 cascade [147]. It should be noted that the seven members of the sirtuin family respond to different stimuli and may even inhibit autophagy given the right circumstances [148]. 

Sirtuin activation has been investigated for clinical application with particular attention to Sirt1 activator resveratrol [149,150,151]. However, resveratrol has several targets, including AMPK, another member of the autophagy induction [152,153]. Some of the other targets of resveratrol may even inhibit autophagy; this has made the adoption of resveratrol as an autophagy inducer difficult [153]. Finally, the sirtuin-enabled LC3 cascade is not strictly necessary for the finalisation of the lysosome. There are documented instance of Atg5–Atg7 independent autophagy that do not involve LC3 [154,155]. Investigations into the effect of UV on Sirt1 found that in cultured cells, a decrease in expression of Sirt1 could be observed alongside a Sirt1-mediated activation of AMPK [155]. Further work in sirtuin-deficient mice found that homozygous knockout animals were sensitized to UVB-induced apoptosis [156]. The roles of sirtuin in controlling autophagy are multiple, but it is certain that sirtuin is necessary in the protection of cells from UV damage. 

While these autophagy mechanisms (see Figure 5) are thoroughly researched, their role in corneal damage repair and particularly their activation following UV damage, is poorly characterised. Autophagy modulating treatments such as resveratrol eye drops are already being researched to address several eye diseases [157,158,159,160], but their exact effect on corneal autophagy in not known. It had already been acknowledged that autophagy likely plays an important role in some UV-induced and UV-propagated diseases such as dry-eye disease and pterygium [161,162]. It has also been theorized that the activation of autophagy and autophagic flux, several pathological conditions in the corneal epithelia and stroma may be prevented [163]. This particular line of inquiry concerns mainly inflammatory conditions but could be applied to UV. 

### 2.7. Apoptosis

Each DNA repair mechanism listed thus far prevents the mutation of a cell into the beginning of cancer. In the event that the extent of the UV damage is too great to be repaired, a cell may undergo apoptosis to ensure the accumulated mutations are not passed on to descendent cells. Unlike necrosis which is characterized by cell swelling and membrane disruption, apoptosis is recognizable by its cell shrinkage and pyknotic nucleus [164]. An apoptotic cell fragments into membrane-enclosed bodies that can be phagocytosed ensures that surrounding cells are mostly unaffected [165]. Apoptosis can be triggered by several factors such as cytokines, hormones, virus, or drugs. At the right concentrations, these stimuli lead to the activation of effector caspases that trigger the apoptotic program [166]. UV irradiation has been shown to cause the activation of ATP-sensitive potassium (K(ATP)) channels in affected cells causing a loss of K+ ions and a depolarization of the mitochondrial membrane [167,168]. Intracellular K+ is importantly linked with the inhibition of caspase, and K+ loss is linked with greater rates of apoptosis [169]. In the cornea specifically, it has been observed that while in vivo cells did not perish following exposure to background doses of UVB, cultured cells did perish when exposed to similar UVB doses [170]. It was found that tear film provided a high amount of K+ ions to the corneal epithelia, ensuring a degree of apoptosis-resistance to the chronically UV-irradiated epithelial cells [171]. This is in contrast with the skin which lacks this excess K+ and instead regularly sees UV-triggered apoptosis in the form of sunburn cells [165,172]. Whether this apoptotic-privilege can be linked to an increased incidence of cancer in the cornea is currently unknown. Importantly, this underlines the relevance of DNA repair mechanisms in the corneal epithelium because epithelial cells are more likely to stay alive, and they need intact DNA to function properly.

## 3. UV Pathogenesis and Rescue

The failure of DNA repair pathways, particularly NER, typically results in developmental disorders, neurological symptoms, and skin cancers. These conditions arise due to the inability of cells to repair DNA in populations of progenitor cells, noncycling cells, and UV-exposed cells [173]. While the severity of NER mutations may vary from solar sensitivity to recurring severe cancers, conditions such as XP, CS and TTD commonly arise from deficiencies in NER. XP can present in multiple ways. It may be accompanied by neurological symptoms as De Sanctis–Cacchione syndrome, it may XP-V be a common variant where only post-replication repair is defective (while this is not a NER defect, because a polymerase is involved, it still results in photosensitivity [174]), or it could present alongside other NER deficiencies such as in the case of XP/CS [175,176]. While the disease itself is variable, one common trait unifies all of them: DNA photolesions are not being repaired.

Among the three main diseases XP, CS, and TTD, while skin-based symptoms are more commonly observed, ocular abnormalities also occur, such as the hallmark retinal degeneration seen in CS [177,178]. Corneal epithelial degeneration and opacities are also observed in CS, with some cases exhibiting rarer symptoms such as accumulation of pigmented macrophages or corneal ulcers [177,179,180]. Chronically UVB exposed CS mouse models developed opaque corneas and bulging eyes. Histology of eyes harvested from these models revealed carcinomas and neovascularization [181]. However, in human CS patients, there are no reports of corneal neoplasm; skin neoplasms are more common [182]. This has been attributed to the greater prevalence of the p48 subunit of the XP-E p48–p125 DNA-binding heterodimer, which is required for enhanced CPD recognition [169,183,184]. XP has been documented as causing neoplastic changes and premalignant melanosis in corneal tissue [185]. The simplest method to prevent DNA photolesions is the avoidance of UV light, be it sunlight or artificial. 

Early work on the failure of UV defense system performed in plants found that a loss of CPD photolyase activity resulted in hypersensitivity to UVB [186]. CPD photolyase does not exist in placental mammals, [187]. There exist animal models with DNA repair mutations that involve the other NER proteins that target CPD and 6-4PP. A model of systemic ERCC1 deficiency showed polyploidy in the liver and kidney [188]. Polyploidy has been positively correlated to cell stress and senescence [189]. A mouse model of ERCC1-knockdown in the corneal endothelium of saw decreased cell density in the corneal endothelium [190]. Unfortunately, there are no reports of loss-of-function NER mutants specific to the corneal epithelia. Studies in humans looking at the NER expression profiles in Fuch’s endothelial corneal dystrophy (FECD) found that XPC was downregulated in patients that had FECD [191,192]. Although this does not make the cells more vulnerable to UV specifically, it does allow for faster accumulation of DNA lesions, which then lead to pathogenesis. 

Because placental mammals lack CPD photolyase due to an evolutionary split, there has been some interest expressing CPD photolyase to potentially provide a much more competent repair enzyme. The photolyase in question is still expressed in marsupials and has been introduced into human cells harvested from an XP patient in an attempt to improve the rate of DNA repair. The marsupial photolyase functioned properly in the host cells and did improve survival of human cells following UV irradiation [193]. Similar experiments that used microinjections of purified photolyases from yeast also found that UV-mediated cell death was reduced, but to a lesser degree compared to the marsupial photolyase [194]. It was hypothesized that the yeast photolyase is not trafficked into the nucleus to the same degree as the marsupial photolyase. With such evidence, work continued with CPD photolyase introduction into placental mammals and a mouse line was generated that ubiquitously expressed either Potorous tridactylus CPD photolyase, Arabidopsis thaliana 6-4PP photolyase, or both [195,196]. The photoreactivation of CPD photolyase resulted in a markedly improved survival of UV-exposed cells in the mice, while photoreactivation of 6-4PPs did not noticeably affect UV-resistance. It should be noted that rodent cells have been documented multiple times as having CPD repair systems less effective than human CPD repair equivalents [197,198,199,200]. This lead to research on the introduction of CPD photolyase into mouse models and showed great improvement in recovery from UV damage [195,196]. 

Further work on repair of CPDs in human cells used an mRNA transfection system to express the marsupial Potorous tridactylus CPD photolyase in UVB-irradiated human epidermal keratinocytes. Results showed greatly reduced CPD amount and reduced activation of UV-induced cytokine such as IL-6 [201]. Use of this system also lead to the identification of which genes where more highly expressed in the presence of CPDs, and thus could be targeted by therapeutic approaches [87]. Cell cycle controllers like cyclin E1 and p15INK4b were expressed in greater quantities following JNK activation by UVB exposure. Several other genes were also identified with UV-dependent expression, although it is uncertain how beneficial the expression of each gene is [202]. Work on the UV-induced CPD-dependent and CPD-independent cellular mechanisms continues, with CPD photolyase as a major tool. The obvious potential of CPD photolyase as a therapeutic agent in humans has been applied, usually with skin in mind [203]. The potential of CPD photolyase in corneal healing comparatively sees less investigation, possibly due to the corneas greater success in dealing with CPDs with existing repair mechanisms or by simply changing the lifestyle of affected patients to better avoid UV exposure. 

## 4. Perspective: Therapeutic Opportunities for UV-Induced Conditions including Pterygium

### 4.1. T4 Endonuclease V and Photolyases as Photolesion-Repairing Treatment Strategies

DNA photolesions play an important role in tumorigenesis and any repair mechanism that repairs these lesions is of great interest in cancer prevention. T4 endonuclease V (T4N5), encoded by the bacteriophage T4 endo V, is an enzyme capable of initiating repair of CPD lesions [204]. If the enzyme can be transported in the nucleus, CPD photolesions could be greatly decreased. A liposome-based delivery system for T4N5 was first developed for delivery into cells and was later applied to skin [205,206]. T4N5 alone can fulfill the role of the multi-complex NER mechanisms in human cells and, in high risk cases where DNA repair mechanisms are faulty like in XP, T4N5 can prevent the development of carcinomas [207]. Certainly, with regards to skin, liposomal T4N5 delivery has had beneficial effects on CPD repair [208]. This has led to the design and distribution of T4N5-containing pharmaceutical products as a protective option for particularly photo-vulnerable patients [209]. The use of photolyase in such products presents a possible continuation of the effort to include not just protective measures but also repairing elements in UV screens. This has been the opinion of previous research groups, and there have already been several medical and cosmetic products developed that incorporate proprietary photolyase blends [210,211,212,213,214]. CPD photolyase derived from blue-green algae was incorporated to protective treatments for human skin [215,216]. Importantly, none of these treatments have been applied to the same extent in cornea; there are no commercially available products and no currently published research that addresses the need for photolesion repair in the cornea. T4N5 and photolyase eye drops could be applied to the cornea for the same purposes of repairing photolesions or protecting particularly vulnerable tissue in NER-deficient or recent transplanted patient.

As with any treatment, and particularly with treatments that affect DNA, there are concerns about toxicity and side-effects. In the case of T4N5, preclinical toxicological tests with single oral doses showed no negative effects [217,218]. Testing on human cells reported that T4N5 had a half-life of 3 h [219]. Further testing on animal eyes with repeated topical application to the cornea found that T4N5 was not an ocular irritant to rabbits or mice and caused no observable histopathological changes [195].

### 4.2. Autophagy-Induction as a Treatment Strategy

There is potential for therapeutic applications of autophagy in treating certain conditions in the eye. Defective autophagy is well documented in corneal diseases such as keratoconus or Fuch’s endothelial corneal dystrophy [220,221]. Rapamycin, a specific inhibitor of mTOR, has the potential to boost autophagy through disinhibition [222]. Initial work with rapamycin showed promise with regards to improvements in corneal epithelial cell survival [223]. Similar work with autophagy inducer trehalose also showed desirable changes in corneal epithelial cells following UV exposure [224] and even accelerated healing in UV-irradiated rabbit corneas [224]. However, boosting autophagy has complex ramifications, and it is possible for treatments such as rapamycin to aggravate stem cell deficiencies [225]. 

## 5. Conclusions

Among all sources of damage to the cornea, UV radiation remains one of the most relevant, with several diseases linked to acute and chronic UV exposure. However, there is no complete understanding of the mechanisms via which damage is repaired and the ways these repair functions can fail. While corneal diseases are rarely deadly and are relatively simple to heal given access to donor material or lifestyle changes, better understanding of the cornea’s UV defenses could be applied to predictive models or even applied to skin and the other UV vulnerable organs. Finally, with a more complete knowledge of how UV damage mitigation might fail, it is possible to conceptualize therapeutic approaches more effectively. 

## Figures and Tables

**Figure 1 biology-11-00278-f001:**
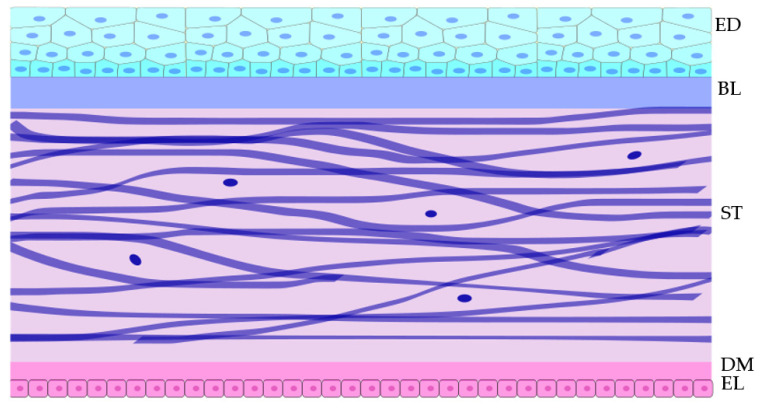
Schematic cross-section of the tissue layers within the central cornea and the approximate shape of cell within the layers. The top cyan layer is the epithelium,;below it is the acellular Bowman layer. Below that in violet are the fibroblasts within the stroma. Below the stroma is the acellular Descemet’s membrane to which the monolayer of pink endothelial cells adhere. Epithelial (ED), Bowman layer (BL), Stroma (ST), Descemet’s membrane (DM), and Endothelial layer (EL).

**Figure 2 biology-11-00278-f002:**
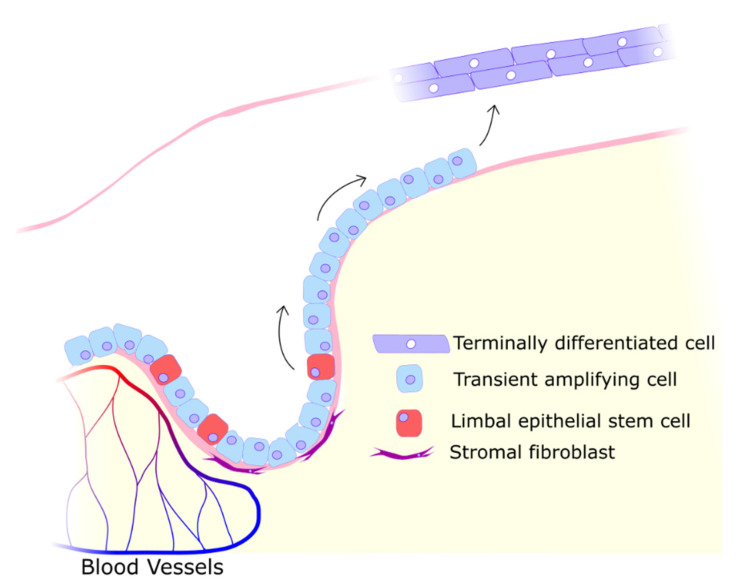
Limbal stem cells differentiating into migratory transient amplifying cells that migrate along the Bowman layer into the avascular central cornea to proliferate and terminally differentiate into functional epithelial cells. The limbal niche is maintained by proximity to blood vessels and secreted factors from stromal cells.

**Figure 3 biology-11-00278-f003:**
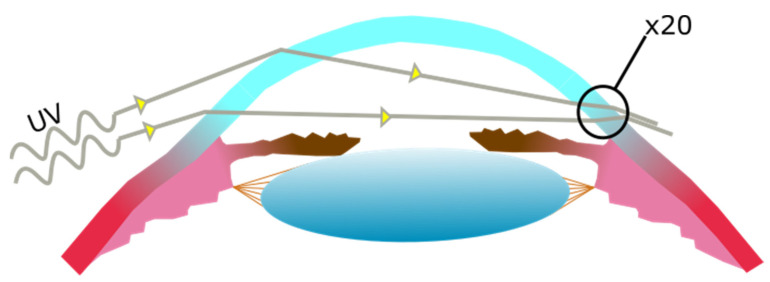
UV light entering the cornea laterally at the temporal side of the cornea, becoming focused at the nasal side of the cornea. The increase in UV exposure at the nasal side is estimated to be of up to twenty times the exposure on the temporal side.

**Figure 4 biology-11-00278-f004:**
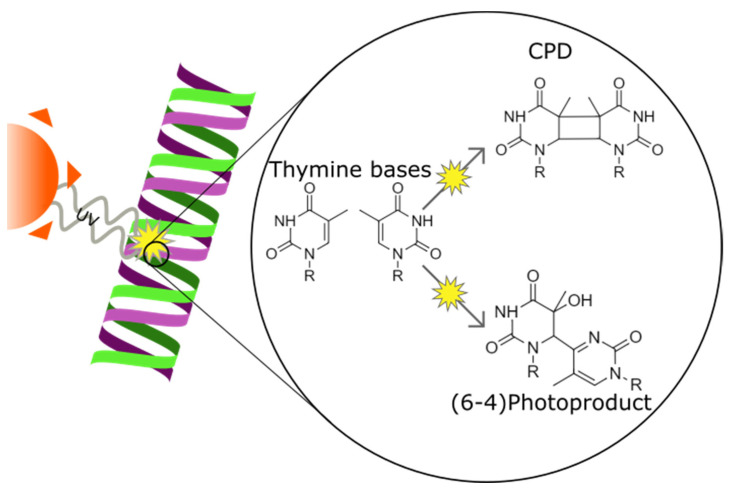
Ultraviolet radiation causing cyclobutane pirymidine dimer and pyrimidine-pyrimidone (6-4) photoproduct formation in a DNA strand between two thymine bases. The two forms of DNA damage differ in the position of the bond created between the two bases.

**Figure 5 biology-11-00278-f005:**
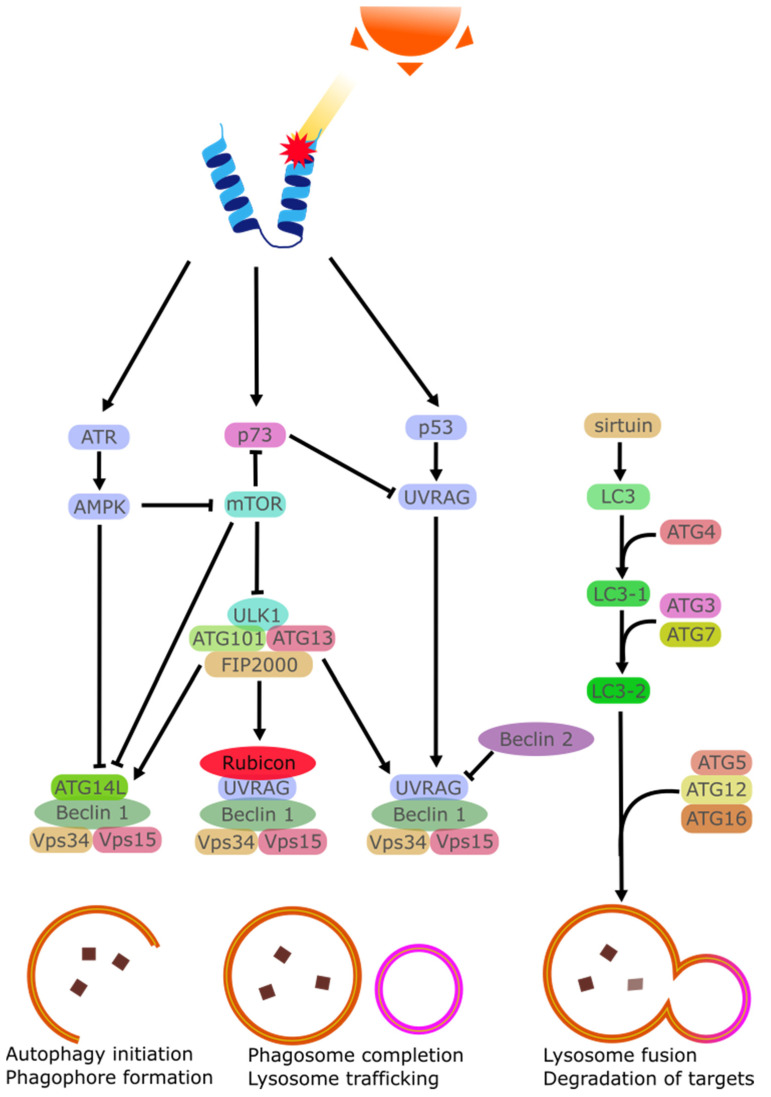
Simplified schematic representation of the autophagic cascade activated by UV damage of proteins.

**Table 1 biology-11-00278-t001:** UV damage and repair mechanisms in both skin and eyes.

Damage	Skin	Cornea	Repair and Prevention
Erythema/Sunburn	✓ [38,39]		Wound Healing, Autophagy [40,41]
DNA lesion	✓ [42,43,44]	✓ [45,46,47]	NER, Apoptosis, Antioxidants [48,49]
Immunodeficiency	✓ [49,50]		None
Premature aging	✓ [51,52]		NER, Apoptosis, Antioxidants [53,54]
Cataracts		✓ [55,56]	Wound Healing, Autophagy [57,58,59,60]
Uveitis		✓ [61,62]	Wound Healing, Autophagy [57,63]
Keratitis		✓ [64,65,66]	Wound Healing, Autophagy [57,67]

## Data Availability

Not applicable.

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
