# Peer review of "UV Protection in the Cornea: Failure and Rescue"

_biology, 2022, doi:10.3390/biology11020278_

Round 1

Reviewer 1 Report

Dear Authors

I enjoyed reading your review which focuses on the UV-protection mechanisms involved in the cornea and skin, the comparison between how these two tissues cope with UV damages, and how these factors could be therapeutically exploited.

I feel the manuscript is well done but could be improved by considering the following comments.

- English must be revised in different parts of the text

i.e.

Line 58: five to six layerS

Line 77: must BE repaired

Line78: EPITHELIAL instead of epithelium

Fig2: secreted factorS

Line 166: papilloma virus IS a source of…

Line 183: proteins ARE

Line 233: CHK2

Line 275: MechanismS are …

Revised the phrase at lines 293-294

Line 297: A NER defect

Line 313: TO prevent

Line 326: DNA lesionS

Line 371: “Later” is repeated

Line 383: treatmentS

Line 387: TRANSPLANTED patient

Line 393: showed promise

Line 406: organS

- IL-6 is not a pathway (line 350)

- Could you describe any possible adverse/collateral effects related to the T4N5 and photolyases treatment strategies?

- Could you specify the differences between GG-NER and TC-NER in terms of when they are differently activated?

- You mentioned BER pathway, could you briefly explain its mechanism of action and the proteins activated to repair ROS-induced lesions?

Good luck with your submission

Author Response

Dear Reviewer,

Thank you for your comments. I have made the following changes based on your suggestions:

-We corrected the spelling mistakes as listed in the reviewer comments

-We added a short paragraph on T4N5 toxicological testing with oral doses, cell cultures, and corneal application, see lines 424-429 below:

As with any treatment, and particularly with treatments that affect DNA, there are con-cerns about toxicity and side-effects. In the case of T4N5, preclinical toxicological tests with single oral doses showed no negative effects [227, 228]. Testing on human cells re-ported that T4N5 had a half-life of 3h [229]. Further testing on animal eyes with repeated topical application to the cornea found that T4N5 was not an ocular irritant to rabbits or mice and caused no observable histopathological changes [230].-Changed "pathway" to "cytokine" in reference to IL-6, see line 362

-Two sentences were added to specify the difference between GG NER and TC NER activation, see lines 147-150 below:

TC-NER and GG-NER differ in the way they recognise DNA damage. GG-NER relies on the XPC complex constantly probing the DNA for lesions while TC-NER relies on RNA polymerase stalling at lesion sites to recognize damaged DNA [212].

-A paragraph was added to explain the detection of ROS-induced DNA damage and how BER detects/repairs these lesions, see lines 213-222 below:

ROS may damage any of the 4 bases of DNA and transform the functional base into any one of several potential lesions [213]. One of these lesions, 8-Oxoguanine (8-OxoG) in particular is linked to greater cancer cell survival [214]. The base excision repair (BER) process that repairs such damage begins with detecting the lesion via DNA glycosylases, in the case of 8-OxoG, Oxoguanine glycosylase 1 (OGG1) binds to the lesion [215]. The DNA glycosylase then cleave the oxidised base out of the DNA, causing a single-strand break. These breaks are detected by PARP proteins that act as the sensor molecules for these single-strand breaks in order to guide gap-filling complexes to repair the break and prevent cell death [216, 217].

Reviewer 2 Report

The review article broadly and fairly fully describes various aspects of UV light-induced DNA damage and its repair and other molecular mechanisms involved in protecting the cornea of the eye. It also presents several therapeutic and perspective options. In my opinion, the review is interesting for a biologist as there is little information about UV mutagenesis and the response to damage in this part of the human body. The manuscript is well written, with perfect original drawings. However, the authors should expand the topic and add a chapter describing some aspects of apoptosis. Besides, I recommend publishing the article in Biology in its current form.

Author Response

We would like to thank the reviewer for their positive review. Per their suggestion, we have added a section that describes the importance of apoptosis in the corneal epitheliumas, how it differs from skin, and how DNA repair is made more important by lower rates of apoptosis. The new section can be found between the lines 302-323, see below:

2.7.Apoptosis

Since each DNA repair mechanism listed thus far prevents the mutation of cell into the begin-ning of cancer. In the event that the extent of the UV damage is too great to be repaired, a cell may undergo apoptosis to ensure the accumulated mutations are not passed on to descendent cells. Unlike necrosis which is characterized by cell swelling and membrane disruption, apoptosis is recognizable by its cell shrinkage and pyknotic nucleus [218]. An apoptotic cell fragment into membrane-enclosed bodies that can be phagocytosed, ensuring that surrounding cells are mostly unaffected [219]. Apoptosis can be triggered by several factors such as cytokines, hormones, virus, or drugs. At the right concentrations, these stimuli lead to the activation of effector caspases that trigger the apoptotic program [220]. UV irradiation has been shown to cause the activation of ATP-sensitive potassium (K(ATP)) channels in affected cells causing a loss of K+ ions and a depolarization of the mitochondrial membrane [221-223]. Intracellular K+ is importantly linked with the inhibition of caspase and K+ loss is linked with greater rates of apoptosis [171]. In the cornea specifically, it has been observed that while in vivo cells did not perish following exposure to background doses of UVB, cultured cells did perish when exposed to similar UVB doses [224]. It was found that tear film provided a high amount of K+ ions to the corneal epithelia, ensuring a degree of apoptosis-resistance to the chronically UV-irradiated epithelial cells [225]. This is in contrast to the skin which lacks this excess K+ and instead regularly sees UV-triggered apoptosis in the form of sunburn cells [219]. Whether this apoptotic-privilege can be linked to an increased incidence of cancer in the cornea is currently unknown. Importantly this underline the relevance of DNA repair mechanisms in the corneal epithelium, since epithe-lial cells are more likely to stay alive, they need intact DNA to function properly.